# Genomic epidemiology of SARS-CoV-2 under an elimination strategy in Hong Kong

Haogao Gu [1,6], Ruopeng Xie[1,2,6], Dillon C. Adam[1,6], Joseph L.-H. Tsui[1,6], Daniel K. Chu[1,6], Lydia D. J. Chang[1], Sammi S. Y. Cheuk[1], Shreya Gurung[1], Pavithra Krishnan[1], Daisy Y. M. Ng[1], Gigi Y. Z. Liu[1], Carrie K. C. Wan[1], Samuel S. M. Cheng[1], Kimberly M. Edwards [1,2], Kathy S. M. Leung [1,3], Joseph T. Wu[1,3], Dominic N. C. Tsang[4], Gabriel M. Leung [1,3], Benjamin J. Cowling [1,3], Malik Peiris [1,2,5], Tommy T. Y. Lam [1,3,5], Vijaykrishna Dhanasekaran [1,2 ✉] & Leo L. M. Poon [1,2,5 ✉]

Hong Kong employed a strategy of intermittent public health and social measures alongside increasingly stringent travel regulations to eliminate domestic SARS-CoV-2 transmission. By analyzing 1899 genome sequences (>18% of confirmed cases) from 23-January-2020 to 26-January-2021, we reveal the effects of fluctuating control measures on the evolution and epidemiology of SARS-CoV-2 lineages in Hong Kong. Despite numerous importations, only three introductions were responsible for 90% of locally-acquired cases. Community outbreaks were caused by novel introductions rather than a resurgence of circulating strains. Thus, local outbreak prevention requires strong border control and community surveillance, especially during periods of less stringent social restriction. Non-adherence to prolonged preventative measures may explain sustained local transmission observed during wave four in late 2020 and early 2021. We also found that, due to a tight transmission bottleneck, transmission of low-frequency single nucleotide variants between hosts is rare.

[1] School of Public Health, LKS Faculty of Medicine, The University of Hong Kong, Hong Kong, China. [2] HKU-Pasteur Research Pole, School of Public Health, LKS Faculty of Medicine, The University of Hong Kong, Hong Kong, China. [3] Laboratory of Data Discovery for Health, Hong Kong Science and Technology Park, Hong Kong, China. [4] Centre for Health Protection, Department of Health, The Government of Hong Kong Special Administrative Region, Hong Kong, China. [5] Centre for Immunology & Infection, Hong Kong Science and Technology Park, Hong Kong, China. [6]These authors contributed equally: Haogao Gu, Ruopeng Xie, Dillon C. Adam, Joseph L.-H. Tsui, Daniel K. Chu. ✉email: veej@hku.hk; llmpoon@hku.hk

Severe acute respiratory coronavirus 2 (SARS-CoV-2) emerged in late 2019[1] and has caused over 170 million confirmed cases and over three million deaths worldwide (as of 1-July-2021)[2]. Heterogeneity in disease severity[3–5] and high virus transmission rates[6,7] necessitated extensive and diverse control strategies, which achieved varied degrees of success. While most countries in Europe and North America adopted suppression strategies to reduce case numbers, other regions including mainland China, New Zealand, and Hong Kong, pursued elimination strategies to prevent importation and community transmission[8]. While countries in Europe and North America reported exponential growth and cocirculation of SARS-CoV-2 lineages with dynamic changes over time and space[9–12], less is known about the dynamics of SARS-CoV-2 in countries that successfully implemented elimination strategies.

Hong Kong (population 7.5 million) has been relatively successful in prevention and control of community SARS-CoV-2 transmission by non-pharmaceutical means, combining intermittent public health and social restrictions, mandatory isolation of cases and quarantine of close contacts in designated facilities[13,14], and increasingly stringent inbound travel regulations to suppress introductions (Fig. 1). As of 1-July-2021, 11,928 laboratory-confirmed cases have been detected, resulting in 211 deaths. Using contact tracing data from January to April 2020 (waves one and two) we have shown that sustained community transmission in Hong Kong was largely driven by superspreading events within social settings, with 80% of community transmission caused by 20% of cases[14]. However, two large community outbreaks have occurred in Hong Kong since this period (waves three and four). Between waves of COVID-19, Hong Kong achieved extended periods of apparently zero community-acquired cases.

Here, we show that during waves two to four, 90% of the confirmed community-acquired SARS-CoV-2 cases in Hong Kong were the result of only three virus introductions (PANGO lineages[15] B.3, B.1.1.63, and B.1.36.27) out of a total of 170 introductions identified through genome sequencing. Using genomic data from travel-related ($n = 186$) and community cases ($n = 1,713$) across all four waves of the pandemic, representing 51.4, 21.1, 23.6, and 13.7% of confirmed cases in each wave, respectively (Supplementary Figs. 1–3 and Supplementary Tables 1 and 2), we discuss the effects of intermittent public health and social measures on the evolution and epidemiology of SARS-CoV-2 in Hong Kong.

## Results

**Introductions, local spread, and detection delays in waves one and two.** During wave one (23-January-2020 to 22-February-2020), laboratory-confirmed cases did not exceed 10 per day (Fig. 1 and Supplementary Table 1) and included both travel-related ($n = 17/70$, 24.3%) and community-acquired cases ($n = 53/70$, 75.7%). The first recognised introduction was detected on 30-January-2020 among a family cluster where the index case had returned from Wuhan, China on 22-January. The first community case of untraceable origin (no travel history or contact with a confirmed case) was also reported on 30-January. However, among two of the earliest locally circulating lineages, time-scaled phylogenetic analysis estimated a median time to most recent common ancestor (tMRCA) of 30-December-2019 (95% Highest Posterior Density Interval (HPD) 24-December-2019 to 1-January-2020), indicating direct ancestry to cases circulating prior to the earliest recognised introductions (Fig. 2a). Critically, these lineages had no recognised epidemiological or phylogenetic link to any other imported cases identified or sampled at that time, indicating that these lineages entered Hong Kong undetected and sustained community transmission during the first wave. Though the tMRCA indicates introduction could have occurred as early as 24-December-2019, which would represent one of the earliest examples of transmission outside mainland China, it cannot be proven conclusively as the lineage diversity observed may have first accumulated in mainland China and subsequently been introduced to Hong Kong closer to the earliest case detections.

As SARS-CoV-2 was declared a global pandemic on 11-March-2020, Hong Kong experienced a substantial rise in travel-related cases ($n = 705/978$, 72.1%, Fig. 1 and Supplementary Table 1) concomitant with large international outbreaks. The majority of wave two introductions came from outside of China (Supplementary Fig. 4), and a moderate increase in community transmission was observed ($n = 273/978$, 27.9%). Again, the inferred common ancestry (tMRCA) of local lineages suggests circulation prior to or during early March 2020, which indicates prolonged patterns of cryptic transmission preceded increases in community spread (Fig. 2a). The largest local lineage, classified as PANGO lineage B.3, was first detected on 18-March-2020 and included 92 genomes or 38% ($n = 92/242$) of all sampled genomes from wave two. This lineage was associated with a superspreading event from which contact tracing identified 106 community cases. Genomic analysis linked an additional 16 sporadic community cases, increasing the total inferred cluster size to 122 or 46.2% ($n = 122/326$) of all community cases during waves one and two.

Overall, during waves one and two, 38 lineages circulated in the community (out of a total of 61 unique importations). The median size for non-singleton local lineages was six sequenced cases, and the median duration of circulation was 10.5 days. Among local lineages without traced contact to an imported case, the median delay in lineage detection (time from tMRCA to first case detection) was 25 days, though this was noticeably higher in January (median = 31 days), before significantly improving to eight days by early May 2020, likely related to early delays and subsequent improvements in case detection (Spearman's test, rho ($\rho$) = −0.49, $p < 0.001$, Fig. 2d).

**Travel measures and the suppression of overseas introductions.** Travel restrictions began as early as 26-January-2020. First, all non-residents that visited Hubei province within two weeks were barred entry into Hong Kong. This was followed by a mandate for compulsory quarantine of passenger arrivals from regions affected with SARS-CoV-2, extending from mainland China to South Korea, Iran, Italy, and the Schengen region, and as imported cases continued to rise, culminated in the barring of entry of non-residents during the peak of wave two in March (Fig. 1 and Supplementary Table 1). Following a peak of community-acquired infections during 16–27 March 2020, control measures such as school closures, adjusted work arrangements, and bans on public gatherings[16] led to a rapid decline. No community transmission was reported from mid-April to mid-May 2020, and community measures were gradually relaxed to allow public gatherings of at most eight (from four) people with restricted opening of leisure venues (Fig. 1). Based on the Oxford COVID-19 Government Response Tracker, which systematically measures variation in government responses[17], stringency of control measures in Hong Kong reduced from level 4 to level 2 during this period (Fig. 1, see "Methods").

**Reintroductions and local surge during waves three and four.** The first prolonged SARS-CoV-2 outbreak in Hong Kong occurred from July to September 2020 (wave three), resulting in >4000 cases (Supplementary Table 1). With a predominant

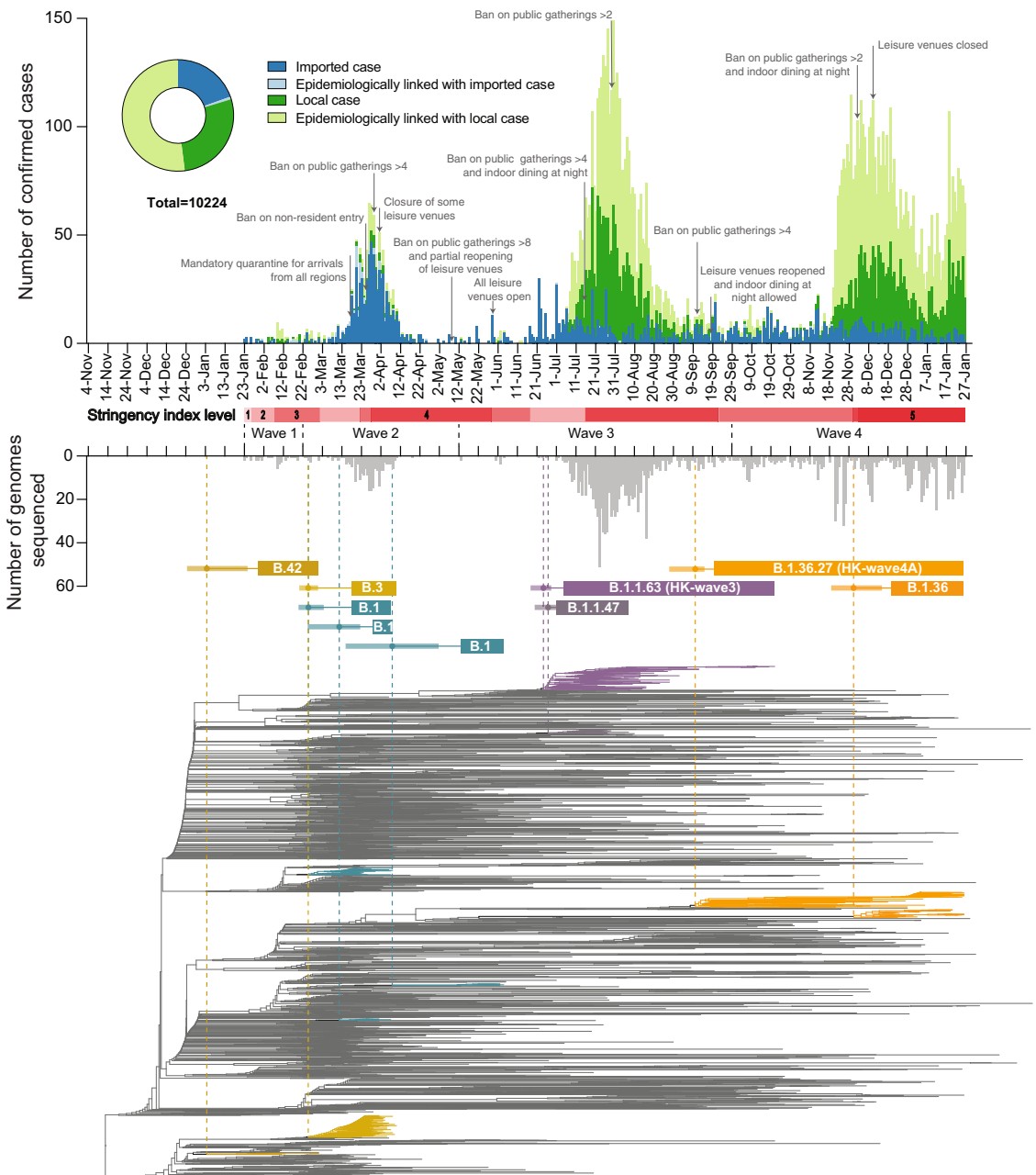

**Fig. 1 Epidemiological summary and time-scaled phylogeny of SARS-CoV-2 in Hong Kong.** Confirmed cases (above) and sequenced genomes (below) are shown as bar charts across the four pandemic waves. Control-measure stringency applied in Hong Kong is based on the Oxford COVID-19 Government Response Tracker[17]. Red shaded bars delineate five levels of control-measure stringency in Hong Kong (Level 1: <40; level 2 : 40-50; level 3: 50-60; level 4: 60-70; level 5: >70). Time-scaled phylogeny of representative genomes from Hong Kong ($n = 610$) and overseas regions ($n = 1,538$) shows monophyletic clades containing at least five community cases in Hong Kong. The two largest Hong Kong lineages during HK-wave3 and HK-wave4A, B.1.1.63 and B.1.36.27, were subsampled to 100 and 65 sequences, respectively. Other PANGO lineages detected during HK-wave3 and HK-wave4A are shown in Supplementary Table 4.

number of cases in the community ($n = 3,385/4,032$, 84.0%), this third wave was preceded by a period of increased detection of travel-related cases during July 2020 (similar to waves one and two, Fig. 1). While importations during wave two were predominantly from Europe, subsequent cases were mostly from Asia (Supplementary Fig. 4, Supplementary Table 3 and Supplementary Data 1). The number of laboratory-confirmed cases continued to rise, reaching a peak of >120 cases per day in late July (Fig. 1). In contrast to waves one and two, community-acquired cases during wave three occurred predominantly among individuals unable to work from home and those not in formal

employment (retired and homemakers)[18] in districts with low income, high density, and other socioeconomic conditions associated with high coronavirus vulnerability[19]. Following the implementation of increasingly stringent public health and social distancing measures, the local epidemic subsided in September. However, beginning in early November a second resurgence (termed wave four) occurred, resulting in >6000 additional cases. Wave four peaked in December 2020 and slowly declined towards zero daily cases by April 2021.

Sequencing identified 170 virus lineages belonging to 71 PANGO lineages in Hong Kong within one year (Fig. 2 and

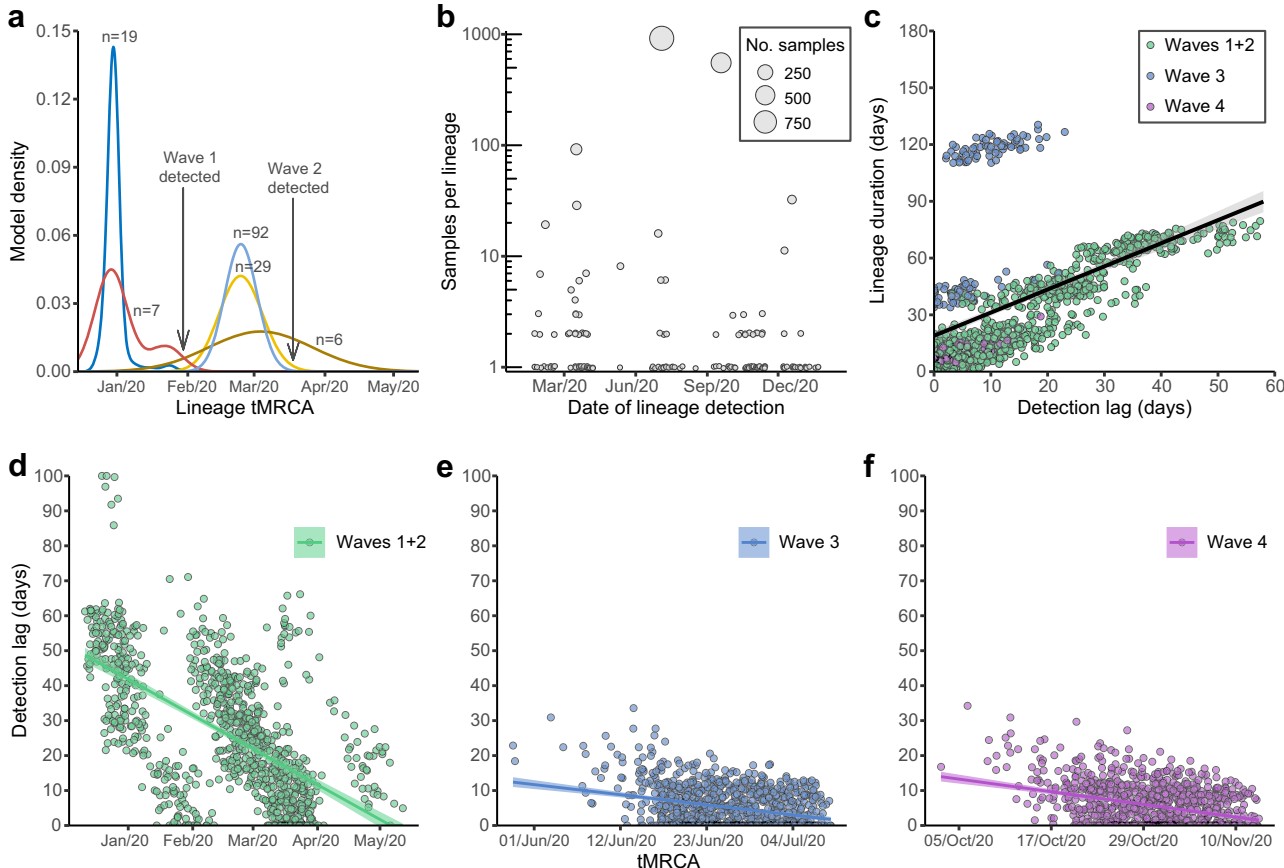

**Fig. 2 Descriptive and temporal dynamics of SARS-CoV-2 lineages in Hong Kong. a** Time to most recent common ancestor (tMRCA) among the five earliest circulating local lineages of SARS-CoV-2 during waves 1 and 2 in Hong Kong. **b** Number of SARS-CoV-2 genomic samples per lineage identified over time using a maximum clade credibility phylogeny. Lineage size is ordered on a log10 scale and plotted by earliest confirmation date. **c** Correlation between the detection lag of locally circulating lineages and the final lineage duration with overlapping points showing uncertainty in lineage detection and duration. Detection lag over time as a function of tMRCA across three epidemic periods **d** waves one and two, **e** wave three, **f** wave four. Overall, a significant reduction in detection lag was observed over time and across each epidemic wave. Points in panels **c**–**f** represent a random sample of 1000 lineages from a Bayesian posterior tree distribution ($n = 8000$).

Supplementary Data 2). However, 87.0% of those lineages were detected only in travel-related cases or single community cases, and no variants of concern were detected in the community during waves three and four. Notably, a single introduction belonging to lineage B.1.1.63 resulted in 92.4% (881/953) of genomes sampled during wave three, forming a HK-wave3 clade that continued to circulate into wave four ($n = 902$ sequences) (Supplementary Fig. 1 and Supplementary Table 4). However, based on the estimated tMRCA of wave three lineages, the earliest imported cases of HK-wave3 were not sampled, indicating cryptic transmission prior to detection. In a similar trend, two lineages led to most of the fourth wave cases: over 74% (552/704) of genomes formed one clade (HK-wave4A, B.1.36.27) and 4.7% (33/704) formed another (HK-wave4B, B.1.36) (Supplementary Table 4).

By comparing the tMRCA of local non-singleton clusters in Hong Kong ($n = 7$, 6.4% (7/109)) (Fig. 2 and Supplementary Table 3), we identified two introductions (HK-wave3 and HK-wave4A) that circulated in the community for 108 and 128 days, respectively. The delay in detection of non-singleton local lineages remained low during waves three and four (mean = 2.9 days, 95% HPD 0–13 days) (Fig. 2e, f), with a detection delay of 11 and 10 days for the two large clades, HK-wave3 and HK-wave4A, respectively (Supplementary Data 3). A negative correlation between delay over time from wave one to wave four (Spearman's

test, rho ($\rho$) = −0.72, $p < 0.001$) reflects improvements in case detection throughout 2020 (Fig. 2e, f). The cryptic circulation of HK-wave3 occurred under relaxed control-measure stringency (level 2). Yet, interestingly, HK-wave4A introduction occurred during the late stages of wave three under stringency level 4 and continued to circulate stealthily as the stringency level was reduced to level 2 on presumed suppression of the wave. Similarly, HK-wave4B introduction occurred under relatively stringent level 4 control measures (Fig. 1). Using a random sample of 1000 posterior trees (each comprising the 170 lineages identified) from a Bayesian tree distribution of 8000 trees in total (see "Methods"), we observed a significant positive correlation between increasing lag in lineage detection and lineage duration (Spearman's test, rho ($\rho$) = 0.70, $p < 0.001$, Fig. 2c).

**Contrasting patterns of epidemiology during waves three and four**. To understand the effects of public health measures on local SARS-CoV-2 transmission, we characterized the two largest clades that circulated during waves three and four and identified contrasting pattens of genomic evolution and underlying transmission dynamics. The effective reproduction number ($R_e$) estimated using a birth-death skyline serial (BDSS) model[20] showed that $R_e$ of HK-wave3 lineage (5-July-2020 to 21-October-2020) was significantly higher (~3) from the tMRCA (mean = 1-July-

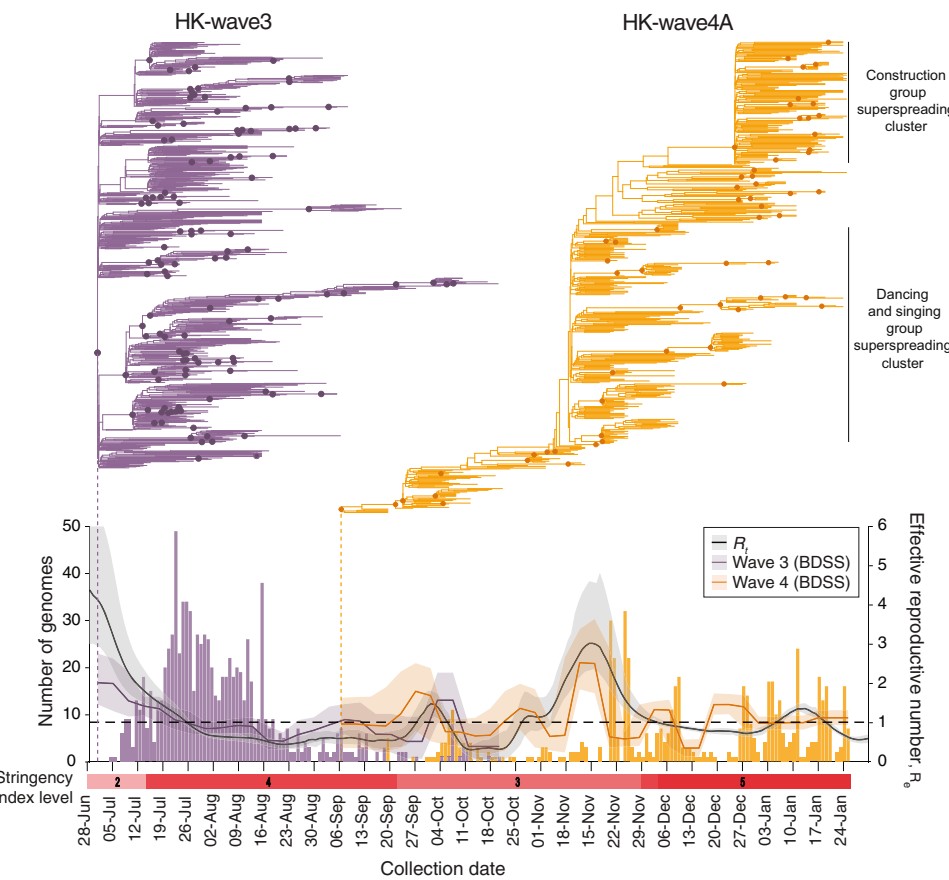

**Fig. 3 Phylodynamics of waves three and four in Hong Kong.** Evolutionary relationships and effective reproduction number ($R_e(t)$) of HK-wave3 (B.1.1.63) and HK-wave4A (B.1.36.27) estimated using tree heights and sequenced incidence data. Node shapes indicate posterior probability >0.5. Histogram shows the number of genomes by collection date. Control-measure stringency applied in Hong Kong is based on the Oxford COVID-19 Government Response Tracker[17]. Black line shows the instantaneous effective reproduction number ($R_t$), estimated based on infection dates inferred from reported symptom onset or confirmation dates for asymptomatic cases.

2020; 95% HPD, 26-June to 4-July) until recognition of the lineage on 5-July when stringency was at level 2 (Fig. 3), highlighting a period of exponential growth as leisure venues reopened and public gatherings of up to 50 people were allowed. However, only 12.6% of HK-wave3 lineage sequences were attributable to social interactions occurring prior to implementation of stringency level 4 (Supplementary Table 5). Although control stringency was intensified on 15-July (from levels 2–4), cases continued to surge over the next two weeks, predominantly among contacts in care homes, households, hospitals, dormitories, and workplaces (Fig. 1 and Supplementary Table 1). However, $R_e$ soon subsided to ~2 and subsequently decreased below ~1 (Fig. 3). Phylogenies reveal a rapid termination of HK-wave3 transmission lineages under level 4 stringency, leading to the disappearance of all but one sub-lineage that continued to circulate with $R_e$ ~1 until extinction in October 2020 (Fig. 3). These results indicate that level 4 stringency during wave three, complimented by aggressive contact tracing, resulted in the elimination of the majority of transmission chains and suppressed virus transmission in social settings.

HK-wave4A (mean tMRCA = 6-September-2020, 95% HPD 5–7 September 2020) continued to circulate for several months despite level 5 stringency (Figs. 1, 3). In contrast to HK-wave3, rapid expansion did not occur upon emergence. Instead, $R_e$ of HK-wave4A fluctuated below 2 throughout September to November (falling below 1 briefly in mid-October), reaching a high of ~3 in mid-November, and fluctuating around 1 in the months that followed. The structure of the HK-wave4A phylogeny suggests

continual elimination of viral lineages with intermittent expansion caused by large superspreading events. The first involved 732 epidemiologically linked cases from 28 dancing and singing venues across Hong Kong beginning in November 2020, and another involved 87 cases linked to three construction sites in January 2021 (Fig. 3). Epidemiological data shows that up to 23.6% of genomes sequenced from HK-wave4A were related to social clusters (Supplementary Table 5), which is significantly higher than the HK-wave3 counterpart ($p < 0.001$, chi-square test). Human mobility levels inferred from Octopus, a smart card payment system used by >98% of the Hong Kong population aged 16–65[21], showed adult and elderly mobility within Hong Kong returned to pre-pandemic levels during the early stage of wave four in early to mid-November 2020 (~100% of the average level during 1–15 January 2020) (Supplementary Fig. 5). Taken together, these results suggest that increased social mixing during a period of relaxed measures, exacerbated by adherence fatigue arising in the population[22,23] due to prolonged social restrictions, likely decreased the probability of lineage termination and sustained community transmission. The instantaneous effective reproduction numbers ($R_t$) of local cases are consistent with $R_e$ of the two major transmission lineages during waves three and four (Fig. 3), indicating the dynamics of waves three and four are indeed driven almost uniquely by HK-wave3 and HK-wave4A, respectively.

**Intra- and inter-host genetic variation.** By analyzing deep-sequence data from confirmed donor and recipient pairs using a

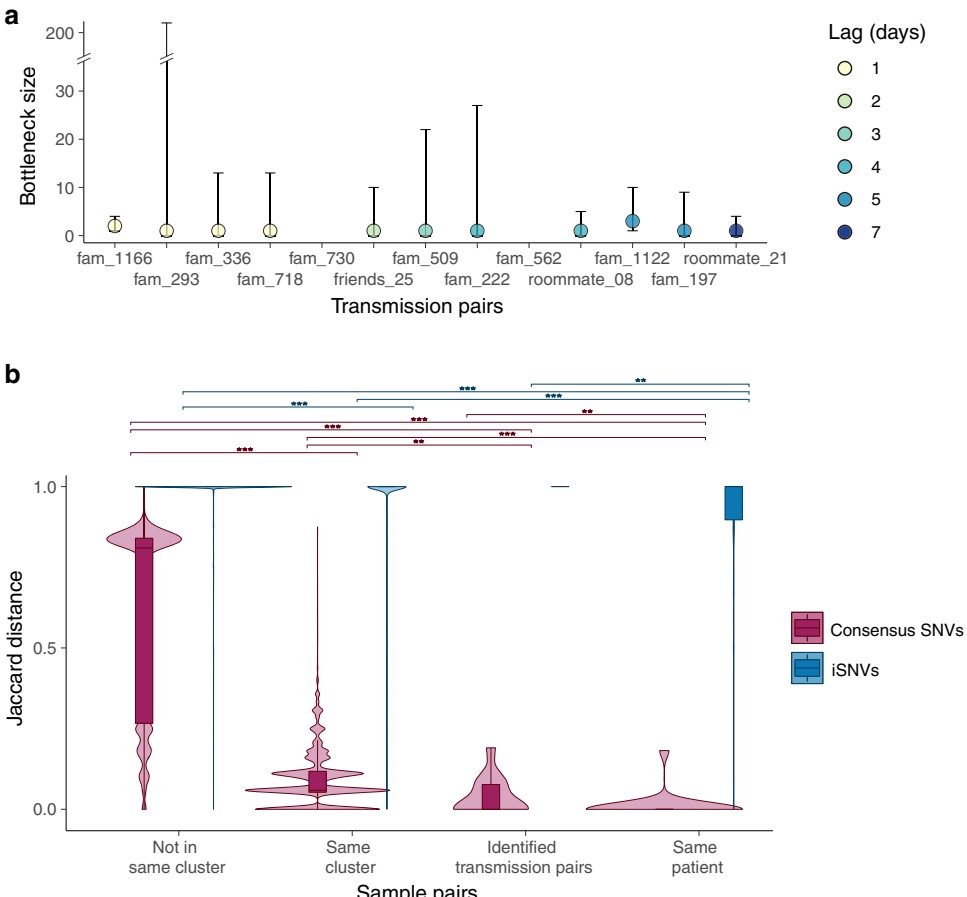

**Fig. 4 Transmission bottleneck size estimates and single nucleotide variant (SNV) frequencies. a** Estimated transmission bottleneck sizes (maximum-likelihood estimates with 95% confidence intervals) for paired donor and recipient samples. Lag time defined by difference in dates of symptom onset. Bottleneck size estimates for transmission pairs fam_562 and fam_730 are not available due to a limited number of intra-host SNVs (iSNVs) in the recipients' samples. **b** Jaccard distance of consensus-level SNVs and iSNVs among epidemiologically and phylogenetically different types of paired samples. Jaccard distances range from 0 to 1, with 0 indicating identical SNV profiles, and 1 indicating no SNVs in common. Violin plots show the range and distribution of Jaccard distances. Boxplots indicate median and inter-quartile ranges (IQR), and whiskers represent value ranges up to 1.5 * IQR. Between-group differences were tested by two-sided Wilcoxon tests separately for consensus-level SNVs and iSNVs. Significance was represented by \*\*$p < 0.05$ ($p = 0.006$ between Same cluster and Identified transmission pairs for consensus-level SNVs; $p = 0.034$ between Identified transmission pairs and Same patient for consensus-level SNVs; $p = 0.014$ between Identified transmission pairs and Same patient for iSNVs) and \*\*\*$p < 0.001$.

beta-binomial statistical framework[24] (see "Methods" for sample selection criteria and controls) we estimated the number of virions required to initiate infection was between one and three (Fig. 4a and Supplementary Table 8). This is consistent with data from transmission pairs estimated in the United Kingdom and Austria (one to eight in the United Kingdom and one to three in Austria[25,26]) as well as between cats (two to five virions[27]), showing the SARS-CoV-2 transmission bottleneck may be universally small.

These results suggest that intra-host single nucleotide variants (iSNVs), defined as variants detected with a minimum depth of 100 reads and minimum frequency >3% but not represented in the sample's consensus genome, are rarely transmitted from donor to recipient host. Comparison of iSNVs using Jaccard distance (see "Methods") identified that iSNVs were dissimilar between patient samples irrespective of epidemiological linkage. Consensus-level SNVs were more similar among transmission pairs (Supplementary Fig. 7) and among samples from epidemiologically and phylogenetically related outbreak clusters (Fig. 4b).

## Discussion

Hong Kong utilized an elimination strategy to control local circulation of COVID-19, yet has so far experienced four distinct

waves. Through genomic sequencing, we were able to investigate the introduction and circulation patterns of SARS-CoV-2 transmission under an elimination strategy[9,28–33]. In contrast to countries with suppression or mitigation strategies[10], where multiple new lineages were reported to cocirculate dynamically, our results show that only two lineages constituted the major viral populations during waves three and four in Hong Kong. Border control measures averted numerous introductions, and community outbreaks were typically associated with exponential growth of virus transmission during less stringent periods and expansion through superspreading events.

Heightened control measures eliminated most domestic transmission chains, but the proportion of social transmission during wave four was significantly higher than that of wave three, indicating control measures were less efficacious when prolonged. Studies characterizing risk perception and protective behaviours in Hong Kong using telephone surveys and mobility data[22,23] estimate a 1.5–5.5% reduction in population compliance during wave four in comparison to wave three, with models estimating a 14% increase in wave four attributable to pandemic fatigue. Comparison of wave three and four strains' ability to replicate in human cells and induce cytokine and chemokine responses[34] suggests that biological differences were

not responsible for the observed variation in infection and transmission dynamics.

Although public health and social measures were promptly lifted when community cases could be traced and controlled, new waves continued as a result of new introductions rather than resurgence of previously circulating viruses. This shows that contact tracing was efficient, but averting outbreaks from new introductions requires heightened border control and enhanced community surveillance during periods of lower control level stringency. Though Hong Kong did not apply highly strict control measures such as city-wide lockdowns or compulsory community testing, as applied in other regions[28,35], the combined use of prompt and proactive contact tracing with mandatory case isolation and contact quarantine requirements, and targeted community measures repeatedly led to the effective elimination of local SARS-CoV-2 transmission over the study period.

This study highlights that continued elimination through rapid implementation of control measures was effective, though the continued prolongation of community measures appeared to reduce their relative effectiveness, likely due to non-adherence-related fatigue. Since suppression of wave four during April 2021, Hong Kong has maintained a 'zero covid strategy' through moderate social restrictions and highly stringent border controls including quarantine on arrival for up to 21 days with intermittent testing and temporary bans on airlines that repeatedly import cases. As a result, Hong Kong has seen only three community cases of 386 total cases from 1-June-2021 to 1-October-2021. In contrast, regions that long maintained elimination strategies, including Singapore, Australia, and New Zealand, have acquired sustained local transmission since mid-2021[36]. Relatively low vaccination rates in Hong Kong compared to high income countries/territories[2] and the emergence of the highly transmissible Delta-variant with partial immune escape[37] complicates the prospect of revising the current elimination strategy.

## Methods

**SARS-CoV-2 data from Hong Kong**. This study was conducted under ethical approval from the Institutional Review Board of the University of Hong Kong (UW 20-168). De-identified saliva or nasopharyngeal samples positive for SARS-CoV-2 by real time-polymerase chain reaction (RT-PCR), along with epidemiological information including onset date, report date, and contact history for individual cases were obtained from the Centre for Health Protection, Hong Kong.

**Genomic sequencing of SARS-CoV-2**. A total of 1753 laboratory-confirmed samples were collected from 1733 RT-PCR confirmed cases from 22-June-2020 to 26-January-2021 (waves three and four). Virus genome was reverse transcribed with primers targeting different regions of the viral genome, published in[38]. The synthesized cDNA was then subjected to multiple overlapping 2 kb PCRs for full-genome amplification. PCR amplicons obtained from the same specimen were pooled and sequenced using Nova sequencing platform (PE150, Illumina). Sequencing library was prepared by Nextera XT (FC-131-1024). The base calling of raw read signal and demultiplexing of reads by different samples were performed using Bcl2Fastq (v2.20, Illumina). A reference-based re-sequencing strategy was applied in analyzing the NGS data. Specifically, the raw FASTQ reads were assembled and mapped to the SARS-CoV-2 reference genome (Wuhan-Hu-1, GenBank: MN908947.3) using BWA-MEM2 (v.2.0pre2)[39]. The consensus sequences for each sample were called as dominant bases at each position by samtools mpileup (v.1.11)[40] with minimum depth of 100 reads. Samples less than 27 kb in length (excluding gaps) were excluded from downstream analysis. The head and tail 100nt bases of all generated consensus sequences were masked. We also masked another 10 sites located in PCR primer binding regions and observed to be variant (≥3% allele frequency) in 1% or more Hong Kong samples (Supplementary Table 9, https://github.com/HKU-SPH-COVID-19-Genomics-Consortium/HK-SARS-CoV-2-genomic-epidemiology). The same masking strategy was also applied in phylodynamic analysis, variant calling, and bottleneck estimation. The average sequencing depth (number of mapped reads) at each nucleotide position that was retained ranged from ~10,000 to ~100,000 (Supplementary Fig. 6).

There were 16 patients from which samples were collected or sequenced at multiple time points. Twelve samples from six patients were sequenced in duplicate, and 21 samples from 10 patients were collected sequentially. One representative sample for each of the 16 patients was selected based on genome

coverage and average sequencing depth. A total of 1899 consensus sequences were included in phylogenetic analysis, including 1601 representative wave three and four sequences that met quality control standards and 298 additional consensus sequences from the first two waves. Sequences from regions outside Hong Kong were retrieved from the GISAID database (total 399,124 sequences, accessed 16-February-2021, detailed accession numbers and acknowledgement information in Supplementary Data 4).

**Phylogenetic analysis of SARS-CoV-2 in Hong Kong**. Hong Kong sequences were analyzed with a global SARS-CoV-2 genome alignment obtained from GISAID (accessed 16-February-2021). For each Hong Kong sequence, the three most similar global sequences (evaluated by $p$ distance excluding gaps, $n = 385$), as well as the earliest sampled sequence ($n = 1279$) from each PANGO lineage (accessed 07-May-2021)[15] were selected. After removing repetitive sequences and trimming masked sites (https://git.io/Jy0eo), data quality was evaluated using a root-to-tip regression analysis in TempEst (v.1.5.3)[41], resulting in a final set of 3437 sequences. Maximum likelihood (ML) phylogenies were estimated using IQ-TREE (v.2)[42], employing the best-fit nucleotide substitution model with Wuhan-Hu-1 (GenBank: MN908947.3) as the outgroup and dated by least square dating (LSD2)[43]. Branch support was estimated using ultrafast bootstrap approximation (UFBoot) and SH-like approximate likelihood ratio test (SH-aLRT), and for nodes of interest with <50% support, we examined their stability through multiple iterative runs using the best-fit nucleotide substitution model. Internal branches with zero-length were preserved for dating by setting parameter $l$ as -1. SARS-CoV-2 sequences from Hong Kong were classified based on the dynamic PANGO nomenclature system (https://github.com/cov-lineages/pangolin, v.2.3.9, 23-April-2021)[15] and confirmed using a ML analysis. Closely related lineages in the community during the early pandemic period, shown in Fig. 2a were delineated based on root branch length and branching pattern of global sequences.

**Phylodynamics of Hong Kong waves**. Monophyletic clusters of SARS-CoV-2 lineages in Hong Kong were determined from the maximum-clade credible (MCC) tree generated using Bayesian molecular clock phylogenetic analysis. Following the pipeline proposed by du Plessis et al.[9] ML trees with branch lengths in genetic distances and time generated in IQ-TREE (v.2)[42] and LSD2[43], respectively were used as inputs for the Bayesian analysis. Time-scaled phylogenies were generated in BEAST (v.1.10) using the strict clock model with 0.001 substitutions per site per year which is within 95% credible interval of SARS-CoV-2 temporal signal[44], the Skygrid model[45] with 61 grid points and a Laplace root-height prior with mean equal to the dated-ML tree estimated by IQ-TREE (v.2)[42] and scale set to 20% of the mean (XMLs can be found in https://git.io/Jy0eX). To improve computational efficiency, the two largest local monophyletic clades in wave three (HK-wave3, $n = 902$) and wave four (HK-wave4A, $n = 552$) from the ML tree were subsampled to 100 and 65 sequences respectively, including the five earliest cases, five latest cases, and 10% of the remainder randomly selected. We ran nine MCMC chains of 100 million, sampling every 1000 steps and discarding 10% as burn-in. As there are no collapsed internal branches in this study, only uncertainty in branch durations was estimated by MCMC. From the approach described in Geoghegan et al.[28], we used the R package "NELSI"[46] to identify all monophyletic lineages, including singletons, and to estimate the delay in lineage detection following importation as well as the duration of circulation, given a set of 8000 posterior trees. It is notable that there are two global sequences from Japan (EPI_ISL_591420 and EPI_ISL_721612) present in the HK-wave3 clade. However, these two cases had travel history to the Philippines (similar to the early HK-wave3 imported cases) and were quarantined upon arrival in Japan, suggesting that they are unlikely to have caused an introduction in Hong Kong. These two cases were therefore excluded when defining the HK-wave3 clade.

For all samples of HK-wave3 and HK-wave4A, we used the birth-death skyline serial (BDSS) model[20] implemented in BEAST (v.2.6.3)[47] to infer the time of origin (tOrigin), time of most recent common ancestor (tMRCA), and temporal variations (piecewise fashion over 12–15 equidistant intervals) in the effective reproductive rate denoted as $R_e$. To estimate $R_e$, a non-informative lognormal prior with a mean (M) of 0 and a variance (s) of 1.0 was chosen. A non-informative prior for tOrigin was used with the lower bound set to 1-January-2020. The HKY + G4 nucleotide substitution model and an uncorrelated relaxed molecular clock model with lognormal rate distribution (UCLN)[48] were used. The sampling proportion was given a uniform distribution as prior with the upper bound on the empirical ratio of the number of sequences to the number of reported cases. MCMC chains were run for 600 million and 800 million steps and sampled every 2000 and 10,000 steps for the lineages HK-wave3 (B.1.1.63) and HK-wave4A (B.1.36.27) respectively, with the initial 10% discarded as burn-in. This resulted in a final total of 270,000 and 72,000 sampled states. Mixing of the MCMC chain was inspected using Tracer (v1.7.1)[49] to ensure an effective sample size (ESS) of >200 for each parameter. Change in the effective reproductive rate ($R_e$) over time after the estimated tMRCA was plotted using R package "bdskytools" (https://github.com/laduplessis/bdskytools). Since by definition there are no sequences between tMRCA and the estimated tOrigin, the $R_e$ was assumed to remain constant in this period. This assumption was incorporated in the default birth-death model using the package TreeSlicer in BEAST (v.2.6.3)[47].

**Human mobility in Hong Kong using Octopus data**. We used digital transactions made on Octopus cards, ubiquitously used by the Hong Kong population for daily public transport and small retail payments (https://www.octopus.com.hk/tc/consumer/index.html), to obtain changes in mobility during 2020–2021 among cards classified as children, students, adults, and elderly (Supplementary Fig. 5).

**Analysis of within-host genetic variation and transmission bottleneck size**. Consensus-level SNVs refer to single-nucleotide mutations present on a sample's consensus sequence in comparison to the reference Wuhan-Hu-1, while iSNVs are defined as variants not present on the sample's consensus sequence (variants from secondary-most alleles, also known as minor alleles) but detected with a minimum depth of 100 reads and at a minimum frequency >3%[26]. SNVs were determined by reference-based alignment of consensus genomes to the Wuhan-Hu-1 reference genome (GenBank: MN908947.3), similar to other SARS-CoV-2 within-host diversity studies[25,50,51]. Reference-based alignment was performed with BWA-MEM2 (v.2.1)[39], and variants were identified using three different variant callers, freebayes (v.1.3.2)[52], VarDict (v.1.82)[53], and LoFreq (v.2.15)[54]. SNVs detected by at least two of the three variant callers were retained for further analysis[55]. Deep sequencing summary statistics are shown in Supplementary Note 1.

We estimated transmission bottleneck size for 13 transmission pairs (donor and recipient), with symptom onset varying by 1 to 7 days. The statistical framework for estimating the transmission bottleneck size between identified transmission pairs was introduced in[24]. It is based on a beta-binomial method that models the number of transmitted virions from donor to recipient. Bottleneck size estimates were calculated by maximum likelihood analysis comparing the allele frequency of variants passing threshold between samples. The 95% confidence intervals were calculated using a likelihood ratio test. Of the 13 epidemiologically linked transmission pairs shown in Fig. 4a and Supplementary Fig. 7, five were sequenced in the same run (fam_1122, fam_1166, fam_336, fam_562, and fam_730), while the other eight pairs were sequenced in different runs.

To identify the similarity of SNV profiles between samples we used the Jaccard distance, defined as one minus the proportion of intersection between two samples divided by the proportion of their union. Jaccard distances ranged from 0 to 1, with 0 indicating identical SNV profiles, and 1 indicating no SNVs in common. Where $A$ and $B$ are two sets of SNVs for comparison:

$$Jaccard(A, B) = 1 - \frac{|A \cap B|}{|A \cup B|}$$

Parameters and scripts for this pipeline are described in https://git.io/Jy0eN. Gene annotations of the SNVs were based on Supplementary Table 10. Between group differences in Fig. 4b were tested by two-tailed two-sample Wilcoxon (Mann–Whitney) tests.

**Estimation of the instantaneous effective reproduction number ($R_t$)**. The instantaneous effective reproduction number ($R_t$) is defined as the average number of secondary cases generated by cases on day $t$. If $R_t > 1$ the epidemic is expanding at time $t$, whereas $R_t < 1$ indicates that the epidemic size is shrinking at time $t$. The transmissibility of imported and local cases was expected to be very different because intensive non-pharmaceutical interventions had been imposed on travellers arriving from COVID-19 affected regions since January 2020. Hence, in the computation of $R_t$, we only included local cases and those epidemiologically linked with local cases as defined by the Centre for Health Protection (CHP, https://www.coronavirus.gov.hk/eng/index.html).

Since the epidemic curves provided by CHP were based on the dates of symptom onset or dates of confirmation, we used a deconvolution-based method to reconstruct the COVID-19 epidemic curves by dates of infection[56,57]. We assumed that the incubation period was Gamma with mean and standard deviation of 6.5 and 2.6 days[58], and that the distribution of the time between symptom onset and case confirmation was Gamma with mean and standard deviation of 4.3 and 3.2 days. For asymptomatic cases, we assumed they shared the same distribution of the time between infection and case confirmation with the symptomatic cases. We then computed $R_t$ for local cases only from the respective epidemic curves using the "EpiEstim"[59] R package (Fig. 3).

**Estimation of the relative reproductive rates of HK-wave3 and HK-wave4A**. We defined the comparative transmissibility of any two lineages as the relative reproductive rate, i.e., the ratio of their basic reproduction numbers. We extended a previous competition transmission model[60,61] of two viruses and applied the fitness inference framework to the sequence data collected in Hong Kong during the cocirculation period of HK-wave3 and HK-wave4A clades (between 19-September and 21-October-2020, Fig. 3). We assumed the two clades shared the same generation time distribution, which can be approximated by the serial interval distribution estimated in Leung et al.[62] (i.e., Gamma distribution with mean and standard deviation of 5.2 and 1.7 days). The inference framework incorporates both incidence and genotype frequency data that reflect the local comparative transmissibility of cocirculating lineages.

**Reporting summary**. Further information on research design is available in the Nature Research Reporting Summary linked to this article.

## Data availability
The Hong Kong SARS-CoV-2 genome sequences and associated metadata generated in this study have been deposited in GenBank (accession numbers are available on GitHub at https://git.io/JyRjK) and GISAID (accession numbers are available on GitHub at https://git.io/JyRD7). Details of confirmed cases of COVID-19 infection in Hong Kong are available from CHP (https://data.gov.hk/en-data/dataset/hk-dh-chpsebcddr-novel-infectious-agent). SARS-CoV-2 reference genome (Wuhan-Hu-1, GenBank: MN908947.3) is available on GenBank. Sequence data from the other countries/regions were obtained from GISAID (accession numbers and acknowledgements are provided in Supplementary Information Data 4). Public transit data was provided by Octopus Cards Limited (Octopus). We obtained consent from Octopus to share the aggregate data of transport transactions between January 1, 2020 and May 31, 2021. Our agreement with Octopus prohibits us from further sharing data with third parties, but interested parties may contact Octopus.

## Code availability
Code used for the above analysis is available on GitHub: https://git.io/JyRyr (https://doi.org/10.5281/zenodo.5797889)[63].

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

## Acknowledgements

We gratefully acknowledge the staff from the originating laboratories responsible for obtaining the specimens and from the submitting laboratories where the genome data were generated and shared via GISAID. We acknowledge the technical support provided by colleagues from the Centre for PanorOmic Sciences of the University of Hong Kong. We also acknowledge the Centre for Health Protection of the Department of Health for providing epidemiological data for the study. The computations were performed using research computing facilities offered by Information Technology Services, the University of Hong Kong. The funding bodies had no role in the design of the study, the collection, analysis, and interpretation of data, or writing of the manuscript. This study was funded by the Health and Medical Research Fund, Food and Health Bureau of the Hong Kong SAR Government COVID190205 (L.L.M.P.), Collaborative Research Fund of the Research Grants Council of the Hong Kong SAR Government C7123-20G (B.J.C.), and the National Institute of Allergy and Infectious Diseases, National Institutes of Health, Department of Health and Human Services of the US, under Contract Nos. U01AI151810 (L.L.M.P.), HHSN272201400006C (V.D., M.P., B.J.C., and L.L.M.P.), and 75N93021C00016 (V.D., M.P., and B.J.C.).

## Author contributions

This study was designed by L.L.M.P. and V.D. Data curation was performed by H.G., R.X., D.C.A., D.N.C.T. and K.M.E. The SARS-CoV-2 genome sequencing team included L.L.M.P., M.P., D.K.C., H.G., L.D.J.C., S.S.Y.C., S.G., P.K., D.Y.M.N., G.Y.Z.L., C.K.C.W. and S.S.M.C. Phylodynamic analysis was performed by V.D., T.T.Y.L., D.C.A., H.G., R.X., K.S.M.L. and J.L.-H.T. Data visualization was done by H.G., R.X., D.C.A., T.T.Y.L. and V.D., L.L.M.P., B.J.C., M.P., T.T.Y.L. and J.T.W. were responsible for project supervision. The original draft of this manuscript was prepared by V.D., R.X., D.C.A., H.G. and T.T.Y.L. and was reviewed and edited by L.L.M.P., V.D., D.C.A., T.T.Y.L., K.M.E., M.P., B.J.C. and G.M.L.

## Competing interests

B.J.C. has consulted for Roche, Sanofi Pasteur, GSK, AstraZeneca, and Moderna. The authors declare no other competing interests.
