## [Peer Review File · Nature Communications]

Genomic epidemiology of SARS-CoV-2 under an elimination strategy in Hong KongREVIEWER COMMENTS

Reviewer #1 (Remarks to the Author):

In this work the authors complement an evolutionary perspective on SARS-CoV-2 evolution in Hong Kong with data on the public health response. This enables them to correlate the efficacy of the latter with the virus's transmission dynamics. Importantly, the results indicate that (i) strict control measures and traveler quarantine do limit the number of successful importations and local transmissions (ii) non-adherence to the prevention measures may explain prolonged sustained local transmission during the most recent wave last year. Both are important observations, and in particular the second major finding is an important flag to raise. However, I believe that the authors should explore alternative explanations for the prolonged persistence of the HK-wave4A lineage when compared to HK-wave3 such as predominant circulation among people with a lower socio-economic status (for which it can reasonably be assumed that it is more difficult to adhere to the prevention measures). From extended data figure 3 it is apparent that the bulk of infections in waves 3 and 4 was in a different district; can factors like population density differences between districts (which in turn may link to socio-economic status) perhaps explain the difference in persistence between HK-wave3 and HK-wave4A? Are there biological differences between the variants responsible for the bulk of wave 3 and 4a infections that can explain the observations - increased transmissibility of the wave 4a variant? At least, alternative explanations should be discussed.

A second major comment is that I interpret the Jaccard distance differently. Specifically, from figure 4 it looks like all differences are statistically significant, but that is not in line with what is mentioned in the text. The author's focus on minor variants is also not in line with the data presented in figure 4b: for the minor variants there appears to not be a link with the infecting genotype, in contrast to what is seen for the major variants. This is opposite to what is written - though this perhaps is a due to an unattentive moment? It could help to include in the methods section a sentence or two about how to interpret the Jaccard dissimilarity index (closer to 1 means more different from each other).

Apart from the above, I have some minor comments/suggestions, which mostly pertain to textual clarifications (see below). Finally, I would like to commend the authors for preparing beautiful figures that nicely complement the text. For these, I only have a few minor suggestions for improvements.

Minor comments

Extended data table: the dates should be rounded to days, and not include info on hours, minutes, seconds. This level of detail is not relevant here.

abstract, line 5: "By analyzing >1700 genome sequences (>17% of confirmed cases)" -> in the methods it is stated that 1601 genomes are analysed. Can be reconciled with each other by rewording to something like 'By analyzing >1700 samples (>17% of confirmed cases)' or alike.

p3 line 28: designed  designated

figure 1: the visibility of the HK clades in the phylogeny can be improved by changing the color of the branches representing non-HK lineages from black to a light-grey. It would also be helpful if the HK-clade names are added to the relevant clade names, in particular HK-wave3, 4A and 4B.

p22 line 41: which sites were masked?

p23 line 56: I don't understand what is meant with "the data tree and starting tree were applied to a ML tree". Can the authors clarify please?

The authors should make the XMLs for the dating analysis available.

p24 line 67: "to identify classify" // typo

p24 line 79: "with a the lower bound " // typo

What prior was used on R_e in the BDSS models? Was this an informative prior or not?

Were the samples of epi-linked people (ext data table 8) sequenced during the same run (to avoid between run variability)? If not, this should be detailed, or a note should be made on why this does/does not affect the conclusions.

Is it correct that the SNVs are called with respect to the sample's own majority rule consensus sequence? It would be helpful if a little more detail on the SNV calling can be given in the methods.

figure 2a: it would be helpful to include an indication of 'wave 1/2' for the tMRCA estimates

p8 line 102: "Following a peak of infections during 16–27 March 2020": from figure 1 it seems that the mandatory quarantine for travelers was introduced coincidental to an increase of imported infections. If correct, this sentence should be reworded to reflect this.

p9 line 136 "However, the earliest imported cases of the HK-wave3 clade were not sampled, indicating cryptic transmission prior to detection" => for clarity it should be detailed what the basis is for this claim.

figure 2c: the circle sizes seem wrong - it now looks like a dozen or so wave 3 lineages were of size 750 (upper left blue circles)?

p10 line 153: it now looks like there are 2 sub-processes: one for lineages with few taxa and one for large lineages. This is worth mentioning if you ask me - unless of course this is a pattern due to a plotting mistake. (see previous remark).

figure 3: it would help interpreting the figure if the stringency level can be co-plotted alike in figure 1. Idem for the relative Octopus mobility intensity over time. The superspreading event cluster should be indicated in the phylogeny in figure 3.

p10 line 166: 'An increasing number of cases continued to be reported' reads weird as in the sentence before the declining R_e over time is mentioned. The next sentence too is not easy to follow. "Phylogenies reveal a rapid termination of transmission lineages among close contacts", yet, it is unclear to what exactly 'close contacts' refer. Also, in this sentence, transmission between close contacts is stopped, while in the final sentence of this paragraph there is mentioning of 'intermittent rise in cases among close contacts'. This should be reworded more carefully.

p11 line 178: I'm not sure that 'propagated' is the correct term here. The next sentence too seems incorrect: is it not inherent that R_e increases when it peaks? Please reword.

Extended Data Fig. 5: it would help if the time period covered by waves 3 and 4 can be indicated on these plots. Also, what quantity do the Y-axes represent?

p13 line 197: "due to fatigue arising in the population" => "due to adherence fatigue arising in the population"?

line 227 "shedding new light on SARS-CoV-2 evolution."  this should be expanded upon: what are these new insights?

line 236: it seems worthwhile to mention that during the less stringent periods there were bursts of transmission (exponential growth phases) instead of just mentioning that there was 'cryptic transmission'.

lines 199-206: R_t is based on all local cases. That it closely mirrors the R_e dynamics estimated from the dominant clades during waves 3 and 4, reassures that these waves' dynamics are indeed driven almost uniquely by these clades. Apart from mentioning this I would not lay focus on the limited differences between the R_t and R_e dynamics - after all this only distracts from the focus of this work.

Reviewer #2 (Remarks to the Author):

This paper illustrates the power of genomic data to understand the epidemiological and transmission dynamics of SARS-CoV-2 in the framework of the controlled outbreaks that occurred in Hong Kong.

The paper is very interesting, detailed and well-written. The reconstruction of the genomic+epidemiological history of the outbreak is undoubtedly very well explained and relevant to understand the impact of different policies and of introductions, and I definitely recommend it for publication.

The only part that I can't understand is the discussion of the major/minor i/SNV and related results at the very end of the Results.

It is not clear at all how these different variants are defined. They can be polymorphic between consensus sequences (SNVs, I imagine) or within samples (minor iSNVs, I imagine), but it is often not clear where the "major SNVs" or "major iSNVs" lie in this classification and how are they actually defined, even after looking at the figures/tables.

Also, within-sample polymorphisms can often result from artefacts, but this is not properly discussed.

A statement that looks especially misleading to me in this context is at lines 225-227: "the SARS-CoV-2 within-host genetic variation is non-random and determined by genomic differences (i.e., consensus sequence)". More clarity would be needed before discarding the possibility of artefacts or transmission of iSNVs, without any implication for the determinants of the generation of intra-host genetic variants.

Comments:

- lines 60-63 and 85-90: could you comment on why you exclude multiple introductions from related viral lineages, e.g. coming from the same local outbreak in China? (See caveat that you discussed in lines 68-70.) Is it just based on the low likelihood of such an event? And what would be the impact on detection delays?
- 196-198: I am not sure I follow where this conclusion comes from. The data presented suggest the opposite, i.e. increased mixing during a post-wave period of relaxation of measures (enhanced by pandemic fatigue, probably) was responsible for the increase in reproduction number.

Responses to Reviewer Comments

General response to reviewers' comments. We thank both reviewers for highlighting the significance of our study and for detailed comments identifying areas for improvement. We have addressed each of the comments below and made corresponding changes in the revised manuscript. Further, to conform to formatting requirements, we reduced the length of the abstract to 150 words and made changes throughout.

Reviewer #1

Reviewer 1. Comment 1. In this work the authors complement an evolutionary perspective on SARS-CoV-2 evolution in Hong Kong with data on the public health response. This enables them to correlate the efficacy of the latter with the virus's transmission dynamics. Importantly, the results indicate that (i) strict control measures and traveller quarantine do limit the number of successful importations and local transmissions (ii) non-adherence to the prevention measures may explain prolonged sustained local transmission during the most recent wave last year. Both are important observations, and in particular the second major finding is an important flag to raise.

However, I believe that the authors should explore alternative explanations for the prolonged persistence of the HK-wave4A lineage when compared to HK-wave3 such as predominant circulation among people with a lower socio-economic status (for which it can reasonably be assumed that it is more difficult to adhere to the prevention measures).

From extended data figure 3 it is apparent that the bulk of infections in waves 3 and 4 was in a different district; can factors like population density differences between districts (which in turn may link to socio-economic status) perhaps explain the difference in persistence between HK-wave3 and HK-wave4A? Are there biological differences between the variants responsible for the bulk of wave 3 and 4a infections that can explain the observations - increased transmissibility of the wave 4a variant? At least, alternative explanations should be discussed.

Response to R1C1.

We made several improvements in explanations for prolonged persistence and modified the abstract to highlight the two major findings delineated by the reviewers.

We refer the reviewer to two separate new studies under peer-review that utilised large-scale weekly telephone surveys and mobility data to characterize changes in risk perception and protective behaviours to COVID-19 in Hong Kong (Du et al. and Liao et al.). Du et al. estimated a 1.5% to 5.5% reduction in population compliance during wave four versus wave three, with modelling suggesting a 14% reduction in wave four if not for pandemic fatigue. Our previous experimental study suggests low biological differences between representatives of wave three and four, with similar replication kinetics in human cells (Chu et al. EID 2021) and lack of differences in cytokine or chemokine induction, suggesting strain differences cannot explain variations in infection and transmission dynamics.

We include the following in Results (lines 125-128) and Discussion (lines 243-251), respectively:

“community-acquired cases during wave three occurred predominantly among individuals unable to work from home and those not in formal employment (retired and homemakers)¹⁸ in districts with low income, high density, and other socioeconomic conditions associated with high coronavirus vulnerability¹⁹”

“Heightened control measures eliminated most domestic transmission chains, but the proportion of social transmission during wave four was significantly higher than that of wave three, indicating control measures were less efficacious when prolonged. Studies characterising risk perception and protective behaviours in Hong Kong using telephone surveys and mobility data^{22,23} estimate a 1.5% to

5.5% reduction in population compliance during wave four in comparison to wave three, with models estimating a 14% increase in wave four attributable to pandemic fatigue. Comparison of wave three and four strains' ability to replicate in human cells and induce cytokine and chemokine responses³⁴ suggests that biological differences were not responsible for observed variation in infection and transmission dynamics.”

Reviewer 1. Comment 2. A second major comment is that I interpret the Jaccard distance differently. Specifically, from figure 4 it looks like all differences are statistically significant, but that is not in line with what is mentioned in the text. The author's focus on minor variants is also not in line with the data presented in figure 4b: for the minor variants there appears to not be a link with the infecting genotype, in contrast to what is seen for the major variants. This is opposite to what is written -though this perhaps is a due to an inattentive moment? It could help to include in the methods section a sentence or two about how to interpret the Jaccard dissimilarity index (closer to 1 means more different from each other).

Response to R1C2. We thank the reviewer for highlighting these lapses in interpretation. We revised this section extensively (Results: lines 217-223, and Methods: lines 385-413) and the main text should be in line with the data presented in Fig. 4b. We have also included interpretation of the Jaccard dissimilarity index as suggested (lines 217-225, 402-404, and legend of Figure 4).

Lines 217-225: “These results suggest that single nucleotide variants (iSNVs) of low frequency, defined as iSNVs detected with a minimum depth of 100 reads and minimum frequency >3% but not represented in the sample's consensus genome, are rarely transmitted from donor to recipient host. Comparison of iSNVs using Jaccard distance (see Methods) identified that low-frequency iSNVs were dissimilar between patient samples irrespective of epidemiological linkage. High-frequency iSNVs were more similar among transmission pairs (Supplementary Fig. 7) and among samples from epidemiologically and phylogenetically related outbreak clusters (Figure 4b). ”

“Jaccard distances ranges from 0 to 1, with 0 indicating identical iSNV profiles, and 1 indicating no iSNVs in common, 1 being identical SNV profiles, and 0.5 indicating maximum variation. Where A and B are two sets of SNVs for comparison: (formula)”

Reviewer 1. Comment 3. Extended data table: the dates should be rounded to days, and not include info on hours, minutes, seconds. This level of detail is not relevant here.

Response to R1C3. Per the reviewer's request, the dates have been rounded to days in the Supporting material.

Reviewer 1. Comment 4. Abstract, line 5: "By analyzing >1700 genome sequences (>17% of confirmed cases)" -> in the methods it is stated that 1601 genomes are analysed. Can be reconciled with each other by rewording to something like 'By analyzing >1700 samples (>17% of confirmed cases)' or alike.

Response to R1C4. The 1601 genomes were in reference to representative sequences from waves three and four, but the study included 1899 genomes in total across all four waves. These sections (lines 3-5 and lines 303-306) were revised for greater accuracy and detail. Please refer to highlighted sentences on.

"By analyzing 1899 genome sequences (>18% of confirmed cases) from 23-January-2020 to 26-January-2021"

"A total of 1,899 consensus sequences were included in phylogenetic analysis, including 1,601 representative wave three and four sequences that met quality control standards and 298 additional consensus sequences from the first two waves.”

Reviewer 1. Comment 5. p3 line 28: designed  designated

Response to R1C5. Modified as suggested (line 28).

Reviewer 1. Comment 6. figure 1: the visibility of the HK clades in the phylogeny can be improved by changing the color of the branches representing non-HK lineages from black to a light-grey. It would also be helpful if the HK-clade names are added to the relevant clade names, in particular HK-wave3, 4A and 4B.

Response to R1C6. Per the reviewer's request, non-HK lineages in Figure 1 are now shown in light-grey.

Reviewer 1. Comment 7. p22 line 41: which sites were masked?

Response to R1C7. We included the masked sites within the GitHub repository (https://github.com/HKU-SPH-COVID-19-Genomics-Consortium/HK-SARS-CoV-2-genomic-epidemiology/blob/master/data/masked_sites_GISIAD.csv), and added a hyperlink to the repository on lines 320-322.

Reviewer 1. Comment 8. p23 line 56: I don't understand what is meant with "the data tree and starting tree were applied to a ML tree". Can the authors clarify please?

Response to R1C8. We clarified these sentences in the revised manuscript (lines 331-333), highlight that we followed the methodology proposed by du Plessis et al., in which ML and dated-ML trees were used as inputs to accelerate BEAST analysis.

"Following the pipeline proposed by du Plessis et al.⁹ ML trees with branch lengths in genetic distances and time generated in IQ-TREE (v.2)⁴² and LSD2⁴³, respectively were used as input for the Bayesian analysis"

Reviewer 1. Comment 9. The authors should make the XMLs for the dating analysis available.

Response to R1C9. Corresponding XML files are now included in the GitHub repository, with references added in Methods (lines 337-339). https://github.com/HKU-SPH-COVID-19-Genomics-Consortium/HK-SARS-CoV-2-genomic-epidemiology/blob/master/data/Phylodynamics/global_sub.xml

Reviewer 1. Comment 10. p24 line 67: "to identify classify" // typo; p24 line 79: "with a the lower bound " // typo

Response to R1C10. Thank you for catching this. The words "classify" and "a" have been deleted (lines 352 and 365).

Reviewer 1. Comment 11. What prior was used on Re in the BDSS models? Was this an informative prior or not?

Response to R1C11. Followed by previous studies (Li N, et al., Serwin K, et al. and Lai A, et al.) on estimation of the effective reproduction number, we used a lognormal distribution $X \sim \text{LogN}(m, s)$ with $m = 0$, $s = 1$, corresponding to a median of 1 (95% interval, 0.193 to 5.18) as an uninformative prior for R_e . We revised this section accordingly on lines 357-358 as follows:

“To estimate R_e , a non-informative lognormal prior with a mean (M) of 0 and a variance (s) of 1.0 was chosen.”

Reviewer 1. Comment 12. Were the samples of epi-linked people (ext data table 8) sequenced during the same run (to avoid between run variability)? If not, this should be detailed, or a note should be made on why this does/does not affect the conclusions.

Response to R1C12. Of the 13 epidemiologically linked transmission pairs, the donor and recipient samples for five pairs (fam_1122, fam_1166, fam_336, fam_562 and fam_730) were sequenced in the same run, while for the other eight sample pairs, the donor and recipient samples were sequenced in different runs. However, our results suggest run variability does not affect the conclusions. Specifically, all the estimated bottleneck sizes ranged from 1-3 irrespective of sequencing run. A Wilcoxon rank sum test showed lack of significant differences between the confidence intervals of the bottleneck size estimation ($p = 0.35$) suggesting little influence of sequencing run on bottleneck size estimation.

We included this information in Methods (lines 398 to 400) as suggested. “Of the 13 epidemiologically linked transmission pairs shown in Figure 4a, the five donor-recipient were sequenced in the same run pairs (fam_1122, fam_1166, fam_336, fam_562 and fam_730), while the remaining eight pairs were sequenced in different runs.”

Reviewer 1. Comment 13. Is it correct that the SNVs are called with respect to the sample's own majority rule consensus sequence? It would be helpful if a little more detail on the SNV calling can be given in the methods.

Response to R1C13. The iSNVs in this study were called basing on the alignment to the Wuhan-Hu-1 reference genome (GenBank: MN908947.3) (Similar to the methods in Lythgoe et al. Science (2021), Kemp et al. Nature (2021) and Wang et al Genome Medicine (2021) etc.). As suggested, we expanded these Methods in lines 380-389.

“High-frequency iSNVs refer to mutations present on the sample's consensus sequence in comparison to the reference Wuhan-Hu-1, while low frequency iSNVs are defined as variants not present on the sample's consensus sequence but detected with a minimum depth of 100 reads and at a minimum frequency $>3\%$ ²⁶. High-frequency iSNVs were determined by reference-based alignment of consensus genomes to the Wuhan-Hu-1 reference genome (GenBank: MN908947.3), similar to other SARS-CoV-2 within-host diversity studies^{25,50,51}. Reference-based alignment was performed with BWA-MEM³⁹, and variants were identified using three different variant callers, freebayes (v.1.3.2)⁵², VarDict (v.1.82)⁵³ and LoFreq (v.2.15)⁵⁴. iSNVs detected by at least two of the three variant callers were retained for further analysis⁵⁵. Deep sequencing summary statistics are shown in Supplementary Note 1.”

Reviewer 1. Comment 14. figure 2a: it would be helpful to include an indication of 'wave 1/2' for the tMRCA estimates.

Response to R1C14. Arrows have been added to indicate the respective dates of Wave 1 and Wave 2 detection relative to the lineage tMRCA estimates.

Reviewer 1. Comment 15. p8 line 102: "Following a peak of infections during 16–27 March 2020": from figure 1 it seems that the mandatory quarantine for travellers was introduced coincidental to an increase of imported infections. If correct, this sentence should be reworded to reflect this.

Response to R1C15. We have revised lines 103-110 as follows:

"Travel restrictions began as early as 26-January-2020. First, all non-residents that visited Hubei province within two weeks were barred entry into Hong Kong. This was followed by a mandate for compulsory quarantine of passenger arrivals from regions affected with SARS-CoV-2, extending from mainland China to South Korea, Iran, Italy, and the Schengen region, and as imported cases continued to rise, culminated in the barring of entry of non-residents during the peak of wave two in March (Figure 1 and Supplementary Table 1). Following a peak of community-acquired infections during 16–27 March 2020, control measures such as school closures, adjusted work arrangements, and bans on public gatherings¹⁷ led to a rapid decline."

Reviewer 1. Comment 16. p9 line 136 "However, the earliest imported cases of the HK-wave3 clade were not sampled, indicating cryptic transmission prior to detection" => for clarity it should be detailed what the basis is for this claim.

Response to R1C16. As suggested, we included a statement to clarify that cryptic circulation was inferred based on the time span from the tMRCA of that lineage to the first case detection.

[lines 139-141]: "based on the estimated tMRCA of wave three lineages, the earliest imported cases of the HK-wave3 clade were not sampled, indicating cryptic transmission prior to detection."

Reviewer 1. Comment 17. figure 2c: the circle sizes seem wrong - it now looks like a dozen or so wave 3 lineages were of size 750 (upper left blue circles)?

p10 line 153: it now looks like there are 2 sub-processes: one for lineages with few taxa and one for large lineages. This is worth mentioning if you ask me - unless of course this is a pattern due to a plotting mistake. (see previous remark).

Response to R1C17. We modified this figure by excluding lineage size in Figure 2c–f. These panels plot a random sample of 1,000 lineages from a Bayesian posterior tree distribution of 8,000 trees. The largest (blue) circles in Fig 2c (suggestive of a separate sub-process) indeed represent the distribution of a large HK-wave3 lineage. We made changes to Figure 2 and the legend to clarify that Fig 2b is based on the mcc tree, while c-f are a sample from a Bayesian posterior tree distribution as described above. We have also removed the different sizes to avoid potential confusion.

Reviewer 1. Comment 18. Figure 3: it would help interpreting the figure if the stringency level can be co-plotted alike in figure 1. Idem for the relative Octopus mobility intensity over time. The superspreading event cluster should be indicated in the phylogeny in figure 3.

Response to R1C18. As suggested, stringency level index was added to Figure 3 and Supplementary Fig. 5. Superspreading clusters (dancing and singing group, construction group) were also labelled in Figure 3.

Reviewer 1. Comment 19. p10 line 166: 'An increasing number of cases continued to be reported' reads weird as in the sentence before the declining R_e over time is mentioned. The next sentence too is not easy to follow. "Phylogenies reveal a rapid termination of transmission lineages among close contacts", yet, it is unclear to what exactly 'close contacts' refer. Also, in this sentence, transmission

between close contacts is stopped, while in the final sentence of this paragraph there is mentioning of 'intermittent rise in cases among close contacts'. This should be reworded more carefully.

Response to R1C19. As suggested, we made several changes to this section to better portray the dynamic changes in cases, stringency and R_e during waves three and four. Please refer to highlighted sentences on p10-11, lines 169-179.

“However, only 12.6% of HK-wave3 lineage sequences were attributable to social interactions occurring prior to implementation of stringency level 4 (Supplementary Table 5). Although control stringency was intensified on 15-July (from level 2 to 4), cases continued to surge over the next two weeks, predominantly among contacts in care homes, households, hospitals, dormitories and workplaces (Figure 1 and Supplementary Table 1). However, R_e soon subsided to ~ 2 and subsequently decreased below ~ 1 (Figure 3). Phylogenies reveal a rapid termination of HK-wave3 transmission lineages under level 4 stringency, leading to disappearance of all but one sub-lineage that continued to circulate with $R_e \sim 1$ until extinction in October 2020 (Figure 3). These results indicate that level 4 stringency during wave three, complimented by aggressive contact tracing, resulted in the elimination of the majority of transmission chains and suppressed virus transmission in social settings.”

Reviewer 1. Comment 20. p11 line 178: I'm not sure that 'propagated' is the correct term here. The next sentence too seems incorrect: is it not inherent that R_e increases when it peaks? Please reword.

Response to R1C20. These lines are modified as follows (lines 182-184):

“ R_e of HK-wave4A fluctuated below 2 throughout September to November (falling below 1 briefly in mid-October), reaching a high of ~ 3 in mid-November, and fluctuating around 1 in the months that followed.”

Reviewer 1. Comment 21. Extended Data Fig. 5: it would help if the time period covered by waves 3 and 4 can be indicated on these plots. Also, what quantity do the Y-axes represent?

Response to R1C21. As suggested, we updated supplementary Fig.5 and added more details on legend as follows:

“Y-axes represent the normalized daily numbers of Octopus transactions according to the average number of Octopus transactions of each age group between January 1, 2020 and January 15, 2020 as benchmark (100%). Shadings indicate pandemic waves in Hong Kong as in Supplementary Figure 1a. (a) 7-day moving average of children, students, adults, and the elderly from January 2020 to June 2021. (b) daily numbers of Octopus transactions from November 2020 to June 2021. (c) The difference of daily numbers of Octopus transactions between seven days.”

Reviewer 1. Comment 22. p13 line 197: "due to fatigue arising in the population" => "due to adherence fatigue arising in the population"?

Response to R1C22. We modified the statement as suggested.

Lines 195-198: “Taken together, our results suggest that increased social mixing during a period of relaxed measures, exacerbated by adherence fatigue arising in the population^{22, 23} due to prolonged social restrictions, likely decreased the probability of lineage termination and sustained community transmission.”

Reviewer 1. Comment 23. line 227 "shedding new light on SARS-CoV-2 evolution."  this should be expanded upon: what are these new insights?

Response to R1C3. Please also see response to Reviewer 1 Comment 2. We have corrected our interpretation of intra-host analysis and the original sentence is removed.

Reviewer 1. Comment 24. line 236: it seems worthwhile to mention that during the less stringent periods there were bursts of transmission (exponential growth phases) instead of just mentioning that there was 'cryptic transmission'.

Response to R1C24. We fully agree with this comment. The sentence has been rewritten. Please refer to highlighted sentence on lines 239-242.

"Border control measures averted numerous introductions, and community outbreaks were typically associated with exponential growth of virus transmission during less stringent periods and expansion through superspreading events."

Reviewer 1. Comment 25. lines 199-206: Rt is based on all local cases. That it closely mirrors the Re dynamics estimated from the dominant clades during waves 3 and 4, reassures that these waves' dynamics are indeed driven almost uniquely by these clades. Apart from mentioning this I would not ley focus on the limited differences between the Rt and Re dynamics - after all this only distracts from the focus of this work.

Response to R1C25. We agree with the reviewer and removed these lines.

Reviewer #2:

General comments. This paper illustrates the power of genomic data to understand the epidemiological and transmission dynamics of SARS-CoV-2 in the framework of the controlled outbreaks that occurred in Hong Kong. The paper is very interesting, detailed and well-written. The reconstruction of the genomic+ epidemiological history of the outbreak is undoubtedly very well explained and relevant to understand the impact of different policies and of introductions, and I definitely recommend it for publication.

We sincerely thank the reviewer for the positive comments on our manuscript.

Reviewer 2. Comment 1.The only part that I can't understand is the discussion of the major/minor i/SNV and related results at the very end of the Results.

It is not clear at all how these different variants are defined. They can be polymorphic between consensus sequences (SNVs, I imagine) or within samples (minor iSNVs, I imagine), but it it often not clear where the "major SNVs" or "major iSNVs" lie in this classification and how are they actually defined, even after looking at the figures/tables.

Also, within-sample polymorphisms can often result from artefacts, but this is not properly discussed. A statement that looks especially misleading to me in this context is at lines 225-227: "the SARS-CoV-2 within-host genetic variation is non-random and determined by genomic differences (i.e., consensus sequence)". More clarity would be needed before discarding the possibility of artefacts or transmission of iSNVs, without any implication for the determinants of the generation of intra-host genetic variants.

Response to R2C1.

We thank the reviewer for pointing out the issue. We have revised these sections and modified Figure 4. We now use iSNV (instead of SNV) throughout to denote intrahost SNV, consistent with other studies. We also changed minor iSNV to “low-frequency iSNV” to avoid any confusion in perceived importance. To clarify these definitions, we revised the Results (lines 217-223), Methods (lines 382-391) and Figure 4 legend, accordingly. Please also see Response to Reviewer 1 Comment 2.

We agree that the similarity of iSNVs profiles between epidemiologically clustered samples may be affected by artefacts or transmission of iSNVs. To avoid sequencing artefact we performed sequencing of five clusters in a single run, and included these comparisons in the Methods. We also avoid over-interpretation and the original sentence was removed. Please see response to Reviewer 1 comment 12.

Reviewer 2. Comment 2. lines 60-63 and 85-90: could you comment on why you exclude multiple introductions from related viral lineages, e.g. coming from the same local outbreak in China? (See caveat that you discussed in lines 68-70.) Is it just based on the low likelihood of such an event? And what would be the impact on detection delays?

Response to R2C2. As identified by the reviewer, we estimated the tmrca separately for closely related lineages in Hong Kong as their root branches were sufficiently long to suggest that they were derived independently. The branching of Hong Kong and non-Hong Kong sequences during this period suggests joint tmrca estimates of closely related lineages will likely represent the tmrca of epidemic activity in the source population in mainland China. We included the following in the methods (lines 327-329) “Closely related lineages in the community during the early pandemic period, shown in Figure 2a were delineated based on root branch length and branching pattern of global sequences.”

Reviewer 2. Comment 3. 195-198: I am not sure I follow where this conclusion comes from. The data presented suggest the opposite, i.e. increased mixing during a post-wave period of relaxation of measures (enhanced by pandemic fatigue, probably) was responsible for the increase in reproduction number.

Response to R2C3. We agree and have modified the text accordingly (lines 185-198).

“Taken together, our results suggest that increased social mixing during a period of relaxed measures, exacerbated by adherence fatigue arising in the population^{22,23} due to prolonged social restrictions, likely decreased the probability of lineage termination and sustained community transmission.”

REVIEWERS' COMMENTS

Reviewer #1 (Remarks to the Author):

I have no further remarks

Reviewer #2 (Remarks to the Author):

The authors have greatly clarified and improved the manuscript. I am fully satisfied with the changes and I think it is a very nice work.

My only advice to improve it: the authors changed their notation to "high-frequency iSNVs" after my comments, but they are not literally iSNVs, as discussed in the Methods. To facilitate interpretation, I would call them something like "SNVs" or "consensus-level SNVs".

Minor typo: "bottlenect" in Figure 4.

Responses to Reviewer Comments

Reviewer #2:

The authors have greatly clarified and improved the manuscript. I am fully satisfied with the changes and I think it is a very nice work.

Comment 1. The authors changed their notation to "high-frequency iSNVs" after my comments, but they are not literally iSNVs, as discussed in the Methods. To facilitate interpretation, I would call them something like "SNVs" or "consensus-level SNVs".

Response. Per reviewers suggestion, we have changed these to "consensus-level SNVs" in Lines 228-232, 419-423, 427, and 448-453 and Figure 4.

Comment 2. Minor typo: "bottleneck" in Figure 4.

Response. Thank you for catching this. We have corrected the typo.